# *Leishmania tarentolae* and *Leishmania infantum* in humans, dogs and cats in the Pelagie archipelago, southern Italy

**Roberta Iatta**[1,2], **Jairo Alfonso Mendoza-Roldan**[1], **Maria Stefania Latrofa**[1], **Antonio Cascio**[3], **Emanuele Brianti**[4], **Marco Pombi**[5], **Simona Gabrielli**[5], **Domenico Otranto**[1,6]*

**1** Department of Veterinary Medicine, University of Bari, Bari, Italy, **2** Department of Interdisciplinary Medicine, University of Bari, Bari, Italy, **3** Infectious and Tropical Diseases Unit, Department of Health Promotion, Mother and Child Care, Internal Medicine and Medical Specialties "G. D'Alessandro", University of Palermo, Palermo, Italy, **4** Dipartimento di Scienze Veterinarie, Università degli Studi di Messina, Messina, Italy, **5** Dipartimento di Sanità Pubblica e Malattie Infettive, "Sapienza" Università di Roma, Rome, Italy, **6** Faculty of Veterinary Sciences, Bu-Ali Sina University, Hamedan, Iran

* domenico.otranto@uniba.it

**Data Availability Statement:** All relevant data are within the manuscript and the representative sequences of pathogens detected in the study were

## Abstract

Visceral leishmaniasis (VL) caused by *Leishmania infantum* is endemic in the Mediterranean basin with most of the infected human patients remaining asymptomatic. Recently, the saurian-associated *Leishmania tarentolae* was detected in human blood donors and in sheltered dogs. The circulation of *L. infantum* and *L. tarentolae* was investigated in humans, dogs and cats living in the Pelagie islands (Sicily, Italy) by multiple serological and molecular testing. Human serum samples (n = 346) were tested to assess the exposure to *L. infantum* by immunofluorescence antibody test (IFAT), enzyme-linked immunosorbent assay (ELISA) and Western blot (WB) and to *L. tarentolae* by IFAT. Meanwhile, sera from dogs (n = 149) and cats (n = 32) were tested for both *Leishmania* species by IFAT and all blood samples, including those of humans, by specific sets of real time-PCR for *L. infantum* and *L. tarentolae*. The agreement between serological tests performed for human samples, and between serological and molecular diagnostic techniques for both human and animal samples were also assessed.

Overall, 41 human samples (11.8%, 95% CI: 8.9–15.7) were positive to *L. infantum* (5.2%, 95% CI: 3.3–8.1), *L. tarentolae* (5.2%, 95% CI: 3.3–8.1) and to both species (1.4%, 95% CI: 0.6–3.3) by serology and/or molecular tests. A good agreement among the serological tests was determined. Both *Leishmania* spp. were serologically and/or molecularly detected in 39.6% dogs and 43.7% cats. In addition to *L. infantum*, also *L. tarentolae* circulates in human and animal populations, raising relevant public health implications. Further studies should investigate the potential beneficial effects of *L. tarentolae* in the protection against *L. infantum* infection.

deposited in the GenBank database with accession numbers MW930736, MW930737 and MW930738.

**Funding:** The authors received no specific funding for this work.

**Competing interests:** The authors have declared that no competing interests exist.

## Author summary

*Leishmania infantum* is the major agent of canine and human leishmaniasis being endemic throughout the Mediterranean basin, including Italy. The protozoan is transmitted through the bite of infected phlebotomine sand flies mainly represented by the species *Phlebotomus perniciosus* and *Phlebotomus perfiliewi* as proven vectors in Italy. The sand fly fauna in this country includes, among others *Sergentomyia minuta*, considered herpetophilic vectors of *Leishmania tarentolae*. This species non-pathogenic to mammalians has recently been detected in human blood donors and in sheltered dogs. Our data demonstrate the occurrence of both *L. infantum* and *L. tarentolae* in humans and animal population living in leishmaniasis endemic area opening new perspectives into the study of this human disease. Therefore, the ecology of *L. tarentolae* highlights the need of a more comprehensive study on the spread of this parasite and on its potential beneficial role in public health through a cross-protection that could occur against pathogenic *Leishmania* spp. when *L. tarentolae* occurs in sympatry.

## Introduction

With over 20 *Leishmania* spp. as causative agents, the leishmaniases are neglected tropical diseases inducing cutaneous, mucocutaneous and, the most severe, visceral form. Visceral leishmaniasis (VL) is mainly caused by *Leishmania donovani* and *Leishmania infantum*, both transmitted by phlebotomine sand fly bites and distributed worldwide, being endemic in over 98 countries across Asia, East Africa, South America and the Mediterranean regions [1]. In particular, *L. infantum* is the most important species of zoonotic concern, with domestic dogs as the main reservoirs and phlebotomine sand flies as vectors. The Mediterranean basin is a hyperendemic area for VL since cases account for up to 6% of the global burden [2]. However, asymptomatic *Leishmania* infections in immunocompetent hosts, are 5–10 times greater than the number of clinically apparent disease cases [3]. Although large outbreaks of VL are uncommon, especially in European countries [4], an epidemic of leishmaniasis started in June 2009, occurred in an urban area of Madrid (Spain) with more than 700 cases recorded [5]. In this scenario, hares (*Lepus granatensis*) were the sylvatic reservoir of *L. infantum*, with a minor role played by dogs [6], and humans presented mainly cutaneous forms, with visceral manifestations in immunocompromised patients [7].

In Italy, VL is typical of rural and peri-urban areas and is present in patchy spots along areas on the Tyrrhenian and the low Adriatic Coasts and on the islands, according to the distribution of the sand fly vectors and of the main dog reservoirs. Sicily, one of the two major Italian islands located in the centre of the Mediterranean Sea, is highly endemic for VL [8,9] along with canine and feline leishmaniosis, given the temperate climate that facilitates the vector circulation and the spread of the parasites [10–13]. A recent surveillance on phlebotomine sand flies conducted in Sicily reported a high abundance of *Sergentomyia minuta* and *Phlebotomus perniciosus*, both competent vectors of *Leishmania tarentolae* and *L. infantum* in reptiles and mammalian hosts, respectively [14]. Unexpectedly, an increasing number of scientific evidence suggests that *S. minuta* feeds also on humans [14–17]. The first evidence of *L. tarentolae* DNA in a human being was from bone marrow and intestines of a 300-year-old Brazilian mummy [18]. Only recently the parasite DNA has been detected in the blood of human donors [17] and in sheltered dogs [16] in central and southern Italy, respectively. In addition, experimental studies demonstrated that *L. tarentolae* may infect human phagocytic cells, differentiate into amastigote-like forms, therefore indicating it may transiently infect human hosts [18,19]. In

addition, this species, non-pathogenic to humans, has aroused great interest in the scientific community given its application as a promising biotechnological expression host and immunotherapy agent for human leishmaniasis treatment [20].

In this context, a comprehensive survey on the occurrence of *L. infantum* and *L. tarentolae* infection in asymptomatic humans, in dogs and cats living in VL endemic area was carried out by performing molecular and serological tests, in order to assess the circulation of these protozoa in a close environment where many interactions among humans, dogs, cats and reptiles occur.

## Methods

### Ethics statement

The study on humans was conducted in accordance with ethical principles (Declaration of Helsinki) and the research protocol was approved by the Ethical Committee of the University Hospital of Palermo, Italy (n. 8/2020). Written informed consent was obtained from each participant.

The animals were handled and sampled following the approval of the Ethical Committee of animal experiments of the Department of Veterinary Medicine, University of Bari, Italy (Approval Number 01/2021).

### Study site

The study was carried out on Lampedusa and Linosa, two small islands of the Pelagie archipelago which is the southernmost Italian territory (205 km from Sicily) and the nearest to North Africa (113 km). The island of Lampedusa (35˚30′56″N 12˚34′23″E) is 20.2 km$^2$ and has a population of 6,000 inhabitants, whereas Linosa is 5.4 km$^2$ with about 300 inhabitants, with a significant increase of population during the summer season due to the arrival of tourists. Lampedusa geologically belongs to the African continent, affecting the vegetation and animal population, whereas Linosa is of volcanic origins and is the closest to Sicily (175 Km). The Pelagie Islands are characterized by a semi-arid and windy climate, moderate rainfall during the mild winters and hot and humid summers, generally with temperature reaching up to 30˚C during the period from April to October.

Data available from Regional Pet Animal Database Sicilian Region (February 2021) has censused a total of 999 dogs (i.e., 794 owned and 205 stray) and 301 cats (i.e., 16 owned and 285 from colony) in both islands belonging to the same municipality.

### Study population and sample collection

From July to October 2020, serum and blood samples of 346 volunteer patients (n = 245 from Lampedusa and n = 101 from Linosa) without any specific sign of leishmaniasis, were originally received from human analysis laboratory for health check analyses (Lampedusa) or purposedly collected (Linosa) under the frame of this study, and subsequently sent to the Parasitology Unit of the Department of Veterinary Medicine, University of Bari (Italy) for serological and molecular testing. During the same period, 99 dogs and 29 cats from Lampedusa and 50 dogs and 3 cats from Linosa were blood sampled. All animals were autochthonous and owned, and an informed consent was signed by the owner before sampling. Before sampling, animals were clinically examined and data (e.g., age, sex, breed, clinical signs) and previous antiparasitic treatments were recorded in each animal's file. Whole blood was collected from each animal in vacuum containers with EDTA (i.e., human = 5mL; dog = 2 mL; cat = 1 mL) and serum collection tubes with clot activator (i.e., human = 5 mL; dog = 2.5 mL; cat = 2.5

mL). Blood and serum samples were tested for *L. infantum* and *L. tarentolae* by molecular and/or serological methods.

## Serological testing

Human serum samples were tested to assess the exposure to *L. infantum* by immunofluorescence antibody test (IFAT), enzyme-linked immunosorbent assay (ELISA) and Western blot (WB) and to *L. tarentolae* by IFAT, whereas canine and feline sera to both *Leishmania* spp. were tested only by IFAT.

The detection of IgG anti-*L. infantum* in humans, dogs and cats was performed as previously described [21–23], whereas for antibodies against *L. tarentolae* in humans, the IFAT was performed by using as antigen whole promastigotes of *L. tarentolae* (strain RTAR/IT/81/ISS21-G.6c/LEM124) following the same procedure for *L. infantum* IFAT using fluoresceinated rabbit anti-human immunoglobulin G (Sigma-Aldrich, Germany) diluted 1:50 as conjugate. Serum samples from each specific host (i.e., human, dog and cat) positive for *L. infantum* by cytological and molecular analyses or WB for human, and a healthy host negative for *L. infantum*, were used as positive and negative controls, respectively in each IFAT. Samples were considered as positive when they produced a clear cytoplasmic and membrane fluorescence of promastigotes from a cut-off dilution of 1:80. Positive sera were titrated by serial dilutions until negative results were obtained.

The detection of antibodies against *L. infantum* in human sera was also performed using a commercial enzyme immunoassay following the manufacturer's instructions (*Leishmania* Ab, Cypress Diagnostics, BE). Briefly, 100µl of serum sample diluted 1:20 in the buffer supplied by the kit, were added to each microwell coated with inactivated *Leishmania* antigens and incubated for 10 min at room temperature. After washing, a second incubation with 100µl of anti-human IgG-conjugate coupled with the horseradish peroxidase was performed. Finally, the substrate was added, and the reaction was blocked with $H_2SO_4$. The absorbance was measured in a microplate reader (Biorad, model 680) at 450 nm. Sera with an absorbance value greater than 1.0 were considered as positive. The human serum samples positive to *L. infantum* by ELISA and/or IFAT were further confirmed by immunoblot assay (*Leishmania* Western Blot IgG, LDBIO Diagnostics, Lyon, FR). A volume of 50µl of serum 1:48 diluted in the buffer supplied by the kit, was distributed to nitrocellulose membranes previously bounded with *L. infantum* antigens. The binding of the alkaline phosphatase-anti human IgG conjugate with the immunocomplex was revealed by the presence on the strip of specific 14 and 16 kDa antigenic bands. Positive and negative controls supplied by the kit were included in both the serological assays.

## Molecular testing

Genomic DNA (gDNA) was extracted from human, dog and cat blood samples by using a commercial GenUPBlood DNA kit (Biotechrabbit GmbH, Hennigsdorf, Germany) respectively, according to the manufacturer's instructions. All samples were tested by duplex real-time PCR (dqPCR) for the detection of partial region of the internal transcribed spacer 1 (ITS1) gene of *L. infantum* and *L. tarentolae* as described in Latrofa et al. [24] and of *L. infantum* kDNA minicircle (120 bp) by real time-PCR (qPCR), using primers, probes and protocol described elsewhere [25]. Genomic DNA from *L. infantum* isolate cultured in Tobie-Evans medium from a leishmaniotic dog living in Italy (zymodeme MON-1) and *L. tarentolae* (strain RTAR/IT/81/ISS21-G.6c/LEM124) promastigotes were used as positive controls, whereas gDNA extracted from blood sample of a healthy dog and negative to *L. infantum* was used as negative control. The DNA samples positive to *Leishmania* by dqPCR were successively

amplified by conventional PCR (cPCR) using primers L5.8S/LITSR targeting partial region of the ITS1 (~300bp) and amplification run as described elsewhere [26]. Amplicons were purified and sequenced in both directions using the same primers as for PCR, employing the Big Dye Terminator v.3.1 chemistry in an automated sequencer (3130 Genetic Analyzer, Applied Biosystems, Foster City, CA, USA). All sequences were aligned using the ClustalW program [27] and compared with those available in GenBank using the BLASTn tool (http://blast.ncbi.nlm.nih.gov/Blast.cgi).

## Statistical analysis

Exact binomial 95% confidence intervals (CI) were established for proportions. The chi-square was used to compare proportions with a probability p value < 0.05 regarded as statistically significant. Agreements between serological tests performed for human samples, and between serological and molecular diagnostic techniques for both human and animal samples were evaluated using Cohen's kappa statistic (κ) as follows: no agreement (κ < 0), slight agreement (0 < κ < 0.20), fair agreement (0.21< κ < 0.40), moderate agreement (0.41 < κ < 0.60), substantial agreement (0.61 < κ < 0.8) and almost perfect agreement (κ > 0.81). Analyses were done using the GraphPad Prism version 8.0.0 (GraphPad Software, San Diego, CA, USA)

## Results

Overall, 41 out of 346 human samples (11.8%, 95% CI: 8.9–15.7) were positive to *L. infantum* (n = 18; 5.2%, 95% CI: 3.3–8.1), *L. tarentolae* (n = 18; 5.2%, 95% CI: 3.3–8.1) and to both species (n = 5; 1.4%, 95% CI: 0.6–3.3) by serology and/or molecular test.

Out of 23 patients positive to *L. infantum* (including 18 + 5 patients exposed only to *L. infantum* and to both *Leishmania* spp., respectively; 6.6%, 95% CI: 4.5–9.8), 17 (4.9%, 95% CI: 3.1–7.7) tested positive by IFAT with antibodies titres of 1:80 (n = 10), 1:160 (n = 4), 1:320 (n = 1) and 1:1280 (n = 2), 21 (6.1%, 95% CI: 4.0–9.1) by ELISA and 16 (4.6%, 95% CI: 2.9–7.4) by WB (**Table 1**). Circulating *L. infantum* kDNA was detected in blood of 2 patients (0.6%, 95% CI: 0.2–2.1) seropositive to both *L. infantum* and *L. tarentolae* with IgG titers of 1:1280 and 1:320, respectively. The two individuals above showed simultaneous positivity to *L. infantum* by all three serological (IFAT, ELISA and WB) and molecular testing (qPCR and/or dqPCR). Kappa agreement between serological tests was almost perfect between WB and ELISA results (κ = 0.86), and substantial between IFAT and WB (κ = 0.78) and IFAT and ELISA (κ = 0.80). Whereas the agreement between different combination of the serological and molecular tests for the detection of both *Leishmania* spp. was slight (k < 0.20).

Out of 23 human samples (6.6%, 95% CI: 4.5–9.8) positive to *L. tarentolae*, 12 (3.5%, 95% CI: 2–6) were positive by IFAT with antibodies titres of 1:80 (n = 6), 1:160 (n = 4), and 1:320 (n = 2) and 11 (3.2%, 95% CI: 1.9–5.6) by dqPCR (**Table 1**).

Out of 13 *Leishmania* spp. dq/qPCR positive samples (i.e., 2 to *L. infantum* and 11 to *L. tarentolae*), 5 specimens were successfully amplified by cPCR targeting the rRNA ITS1 gene,

**Table 1. Number and prevalence of humans exposed to or infected by *L. infantum* (*Li*) and *L. tarentolae* (*Lt*) based on the serological and molecular diagnostic tests and geographical origin.**

| Test | IFAT | | ELISA | WB | dq/qPCR | |
|---|---|---|---|---|---|---|
| | *Li* | *Lt* | *Li* | *Li* | *Li* | *Lt* |
| Lampedusa (n = 245) | 10 (4.1%) | 10 (4.1%) | 12 (4.9%) | 8 (3.3%) | 2 (0.8%) | 7 (2.9%) |
| Linosa (n = 101) | 7 (6.9%) | 2 (2.0%) | 9 (8.9%) | 8 (6.9%) | 0 | 4 (4.0%) |
| **TOT** (n = 346) | 17 (4.9%) | 12 (3.5%) | 21 (6.1%) | 16 (4.6%) | 2 (0.6%) | 11 (3.2%) |

**Table 2. Number and prevalence of dogs positive to *L. infantum* (*Li*) and/or *L. tarentolae* (*Lt*) by IFAT and qPCR based on the geographical origin.**

| Test | IFAT | | | qPCR |
|---|---|---|---|---|
| | *Li* | *Lt* | *Li + Lt* | *Li* |
| Lampedusa (n = 99) | 19 (19.2%) | 2 (2.0%) | 32 (32.3%) | 1 (1.0%) |
| Linosa (n = 50) | 4 (8.0%) | 0 | 2 (4.0%) | 1 (2.0%) |
| **TOT** (n = 149) | 23 (15.4%) | 2 (1.3%) | 34 (22.8%) | 2 (1.3%) |

confirming *L. tarentolae* (n = 4) and *L. infantum* (n = 1) species identification with a nucleotide identity of 99–100% with the *L. tarentolae* reference sequence MT416142 and of 100% with *L. infantum* sequence MN648764 available in the GenBank database. Representative sequences of pathogens detected in this study were deposited in the GenBank database (*L. tarentolae* 278bp, 276bp with accession numbers MW930736, MW930737; *L. infantum* 265bp with accession number MW930738).

None of the individuals (i.e., 162 males and 184 females; median age, 49 years; age range, 21 to 87 years) included in the study had a medical history of visceral or cutaneous leishmaniasis.

Out of 149 dogs, 59 (39.6%; 95% CI: 32.1–47.6) were exposed to *Leishmania* spp. of which two, seropositive to both *Leishmania* spp., scored positive by qPCR to *L. infantum* (**Table 2**).

Twenty-three seropositive dogs showed antibodies titers against *L. infantum* by IFAT of 1:80 (n = 12), 1:160 (n = 6), 1:320 (n = 4) and 1:640 (n = 1). Two dogs were seropositive to anti-*L. tarentolae* with titers of 1:80 and 1:160. Thirty-four dogs tested positive to IgG against both species, had titers ranging from 1:160 to 1:2560 for *L. infantum* and from 1:80 to 1:1280 for *L. tarentolae*. Two animals positive to *L. infantum* DNA by qPCR were also seropositive to both species. The prevalence of dogs living in Lampedusa and exposed only to *L. infantum* or *L. tarentolae* or to both species (53.5%, 95% CI: 43.8–63) was statistically higher than that recorded in those from Linosa (12%, 95% CI: 5.6–23.8) ($\chi2$ = 23.9, df = 1, p < 0.00001).

As far as cats from Lampedusa, 14 out of 32 (43.7%, 95% CI: 28.2–60.7) were positive to *L. infantum* (n = 4; 12.5%, CI: 5–28.1), *L. tarentolae* (N = 3; 9.4%, 95% CI: 3.2–24.2) and to both species (n = 6; 18.2%, 95% CI: 8.9–35.3) by IFAT and one seronegative was positive to *L. infantum* (3.1%, 95% CI: 0.5–1.6) by qPCR. Ten cats seropositive to *L. infantum* had antibodies titers of 1:80 (n = 7) and 1:160 (n = 3), whereas 9 seropositive to *L. tarentolae* of 1:80 (n = 7) and 1:160 (n = 2). The three cats from Linosa scored negative by IFAT as well as by dp/qPCR. The K agreement between IFAT and the molecular tests in dogs and cats was slight (k < 0.20).

## Discussion

Findings of this survey indicate that either *L. infantum* and/or *L. tarentolae* may infect humans living in the Pelagie archipelago with the same prevalence (i.e., 6.6%). Although asymptomatic *L. infantum* infection is expected in a geographical area highly endemic for VL, the detection of antibodies against *L. tarentolae* and of circulating DNA in the blood samples of the screened population is of major interest and opens new perspectives into the study of human leishmaniasis. Indeed, the presence of specific antibodies anti-*L. infantum* revealed by IFAT, ELISA and WB in 23 patients (6.6%) and the detection of circulating *L. infantum* kDNA (0.6%) indicates an important cumulative exposure to or infection by the parasite in the examined populations in the investigated area. Meanwhile, the low IgG titers (up to 1:320) against *L. infantum* detected by IFAT in patients molecularly negative may suggest an exposure to the pathogen. Contrarily, the detection of *L. infantum* DNA only in two patients with high antibody titers (1:1280) may suggest the occurrence of subclinical infection or a recent exposure to the protozoan infection.

Though the prevalence of *L. infantum* infection in asymptomatic humans, including blood donors, is variable (0.6–71%), depending on the diagnostic test and the geographical area [28], prevalence of positivity herein recorded are overall consistent with those recorded in healthy subjects in other European countries such as southern France [29], southern Spain [30], the Balearic Islands [31] and Italy [32,33]. In endemic areas, most *Leishmania* spp. infections remain asymptomatic, indeed the ratio of subclinical to clinical cases of VL has been estimated to be up to 50:1 based on parasite virulence and host susceptibility [34–36]. Asymptomatically infected hosts can harbor viable parasites throughout life and may develop the clinical disease if immunosuppression occurs thereafter [37]. Moreover, the parasite transmission may be maintained by asymptomatically infected hosts exposed to bites of competent sand fly species [38]. The comparison of the serological tests for the detection of antibodies against *L. infantum* revealed a substantial agreement between them, with a slightly higher score among ELISA and WB suggesting a higher specificity of these tests than IFAT. Therefore, IFAT and ELISA are recommended for routine screening while WB as confirmatory test in the diagnosis of VL [33]. The slight agreement between serological and molecular tests for the detection of anti-bodies or DNA of both *Leishmania* spp. was expected because of the smaller number of molecularly positive samples from humans than those seropositive.

Strikingly, the human population under investigation turned out to be also exposed to or infected by *L. tarentolae*, which has been historically considered a saurian-associated *Leishmania* spp.. This data is of importance considering that the serological positivity (i.e., 3.2%) has been reported herein for the first time. Such seropositivity overlaps that recorded by molecular testing (3.1%). The first evidence of *L. tarentolae* DNA in humans was in soft and hard tissues, including bone marrow, of a Brazilian mummy dating to the end of 18[th] century [18], and recently in blood donors from central Italy [17]. Both findings suggest the occurrence of parasite systemic spreading in humans. In addition, *L. tarentolae* is experimentally capable of infecting human phagocytic cells and differentiating into amastigote-like forms [19]. There-fore, these findings along with our results suggest that *L. tarentolae* may naturally infect and circulate in human hosts. Indeed, *L. tarentolae* is commonly regarded infecting geckoes (i.e., *Tarentola annularis* and *T. mauritanica*) and recently also lacertid lizards (*Podarcis siculus*) [16]. Nonetheless, some strain (i.e., LEM-125) of *L. tarentolae* and other species belonging to the subgenus *Sauroleishmania*, such as *Leishmania adleri*, may cause transient infections in humans [19,39].

Although the accuracy of the IFAT for the detection of antibodies anti-*L. tarentolae* has not been investigated and cross-reactions with highly similar species of *Leishmania* may occur, the presence of *L. tarentolae* DNA in the blood of 11 patients suggests that this protozoan may infect humans. Therefore, the sensitivity and specificity of this serological method, should be evaluated by comparing it with a more accurate test (i.e., WB) as reported for the diagnosis of VL.

To date, *L. tarentolae* is considered a promising parasite for the expression of human recombinant proteins [20,40], immunotherapy agent for human leishmaniasis treatment as demonstrated in the murine model [41] and vaccine candidate [42], thus suggesting a potential protective and beneficial effect against *L. infantum* in hosts naturally infected by the parasite. Moreover, considering that *L. tarentolae* DNA has been detected in *P. perniciosus* and *Phlebo-tomus perfiliewi*, which usually feed on mammals, as well as in *S. minuta* fed on humans [16,17], the epidemiology of this parasite requires careful and thorough evaluation.

As far as the dog and cat populations living on the Islands, the overall exposure to *L. infan-tum* is high in both dogs (37.6%) and cats (28.1%), with higher seroprevalence (i.e., 51.5%) and antibody titers in dogs from Lampedusa than from Linosa. The higher statistically significant prevalence of seropositive dogs from Lampedusa (53.5%) than Linosa (12%) may be due

mainly to the rainy and windy conditions in the latter island which are unfavorable for phlebotomine sand flies. Overall, data overlap those of a previous survey where more than 50% of dogs scored positive to the parasite and where a high abundance of *P. perniciosus* (67.7%) and *S. minuta* (28.5%) was detected [10]. A similar proportion of canine (41.8%) and feline (25.7%) infection by *L. infantum* was reported in a study conducted on the Aeolian Islands (Sicily) [11], demonstrating that humans and animal populations, in such confined environment, have a high risk of *Leishmania* infection.

The positivity of dogs to *L. tarentolae* by serological (24.2%) and molecular testing (1.3%), was recently reported in shelter dogs (5% and 1%, respectively) in southern Italy where a high incidence of *L. infantum* subclinical infection (10%) was also recorded along with a high abundance of *S. minuta* [16]. These findings along with the serological positivity to *L. tarentolae*, firstly retrieved also in cats, suggest the circulation of both *Leishmania* species in human and animal populations.

## Conclusions

Humans, dogs, reptiles, and sand flies living in the same confined environment in small islands in the middle of the Mediterranean Sea may share *Leishmania* spp. including the non-pathogenic *L. tarentolae*. The presence of *L. tarentolae* in humans and dogs, undoubtedly raises many scientific questions about the potential beneficial effects this species of *Leishmania* may have in cross-protecting hosts infected by pathogenic *Leishmania* species. Under the above circumstances, more comprehensive studies on the occurrence of this parasite need to be addressed and its usefulness in public health evaluated. For example, the expression of recombinant antigens in live vectors (e.g., *L. tarentolae*) has been investigated for its protective effect for preparing vaccines against *L. infantum* [20,42]. As future perspectives, studies focus on the validation of more accurate tests (e.g., WB) for a proper serological diagnosis of *L. tarentolae* infection in humans and animals along with experimental infection studies in mammals would be advised to establish whether the parasite effectively replicates in macrophages. This could eventually provide important knowledge on the infection establishment in the host and in the stimulation of the animal immune response.

## Acknowledgments

The Authors would like to thank Natale Sergio Glorioso (ASP Palermo, Dipartimento di Prevenzione Veterinario, U.O.S.D., Anagrafe animale), Tommaso Lombardo (Direzione ASP, Palermo), Salvo Sotera (Veterinary practitioner, Lampedusa), Irene Cambera (Marine biologist, Linosa) and Sara Tuccio for their support during the field activities and the data collection and Lorena Zecca (Dipartimento di Sanità Pubblica e Malattie Infettive, "Sapienza" Università di Roma) for the serological analysis.

## Author Contributions

**Conceptualization:** Roberta Iatta, Jairo Alfonso Mendoza-Roldan, Domenico Otranto.

**Data curation:** Roberta Iatta, Jairo Alfonso Mendoza-Roldan, Maria Stefania Latrofa, Simona Gabrielli, Domenico Otranto.

**Formal analysis:** Roberta Iatta, Jairo Alfonso Mendoza-Roldan, Maria Stefania Latrofa, Simona Gabrielli, Domenico Otranto.

**Investigation:** Roberta Iatta, Jairo Alfonso Mendoza-Roldan, Simona Gabrielli, Domenico Otranto.

 *Leishmania tarentolae* and *Leishmania infantum* in humans and animal populations

**Methodology:** Roberta Iatta, Jairo Alfonso Mendoza-Roldan, Maria Stefania Latrofa, Simona Gabrielli, Domenico Otranto.

**Project administration:** Roberta Iatta, Jairo Alfonso Mendoza-Roldan, Domenico Otranto.

**Supervision:** Domenico Otranto.

**Validation:** Roberta Iatta, Jairo Alfonso Mendoza-Roldan, Maria Stefania Latrofa, Simona Gabrielli, Domenico Otranto.

**Visualization:** Roberta Iatta, Jairo Alfonso Mendoza-Roldan, Maria Stefania Latrofa, Domenico Otranto.

**Writing – original draft:** Roberta Iatta, Domenico Otranto.

**Writing – review & editing:** Roberta Iatta, Jairo Alfonso Mendoza-Roldan, Maria Stefania Latrofa, Antonio Cascio, Emanuele Brianti, Marco Pombi, Simona Gabrielli, Domenico Otranto.

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
