## [Decision Letter · Decision Letter 0]

24 Aug 2021

Dear Dr. Otranto,

Thank you very much for submitting your manuscript "Leishmania tarentolae and Leishmania infantum in humans, dogs and cats in the Pelagie archipelago, southern Italy" for consideration at PLOS Neglected Tropical Diseases. Apologies for the delay in sending a decision, but we had a delay finding suitable reviewers for your work. As with all papers reviewed by the journal, your manuscript was reviewed by members of the editorial board and by several independent reviewers. In light of the reviews (below this email), we would like to invite the resubmission of a significantly-revised version that takes into account the reviewers' comments. 

We cannot make any decision about publication until we have seen the revised manuscript and your response to the reviewers' comments. Your revised manuscript is also likely to be sent to reviewers for further evaluation.

Sincerely,

Yara M. Traub-Csekö

Associate Editor

Alvaro Acosta-Serrano

Deputy Editor

Reviewer's Responses to Questions

**Key Review Criteria Required for Acceptance?**

**Methods**

-Are the objectives of the study clearly articulated with a clear testable hypothesis stated?

-Is the study design appropriate to address the stated objectives?

-Is the population clearly described and appropriate for the hypothesis being tested?

-Is the sample size sufficient to ensure adequate power to address the hypothesis being tested?

-Were correct statistical analysis used to support conclusions?

-Are there concerns about ethical or regulatory requirements being met?

Reviewer #1: All clear.

Reviewer #2: It is not clear how the sizes of human, canine and feline populations were determined.

Reviewer #3: In this manuscript authors show interesting results of a potentially new Leishmania transmission scenario, involving a parasite from the subgenus Sauroleishmania. However, in order to strengthen the findings, more robust methods should be used. 

Some relevant questions:

Can you perform an identification of the L. tarentolae strain (TAR and LEM) to provide further evidence of the infection potential to human hosts?

How specific are the serological tests used for L. tarentolae screening? 

Can you include other Leishmania species in your IFATS to account for inespecificities?

**Results**

-Does the analysis presented match the analysis plan?

-Are the results clearly and completely presented?

-Are the figures (Tables, Images) of sufficient quality for clarity?

Reviewer #1: Lines 208-210: were these the same samples? meaning, were the serologically positive individuals also PCR-positive?

lines 217-219: please, also give the number of base pairs (lengths of the respective fragments)

line 240: why 53.5%? how does this correlate to data given in table 2?

Reviewer #2: Comparison between serological and molecular results could have been made.

Reviewer #3: ITS based analyses are characterized by a short fragment, which can lead to inespecificities. To strengthen the results, authors could include a marker with a longer fragment for the detection of L. tarentolae. In this way, they could perform comparative analysis, even including other Leishmania species. HSP70 has already been used in L. tarentolae and it generates larger fragments, so there are good reference bases. Also including a tree would be very helpful. 

Regarding serological tests as the basis for the diagnosis of these two Leishmania species, the specificity of the test against L. tarentolae is not clear, again, the use of other Leishmania species as reference would be desirable.

**Conclusions**

-Are the conclusions supported by the data presented?

-Are the limitations of analysis clearly described?

-Do the authors discuss how these data can be helpful to advance our understanding of the topic under study?

-Is public health relevance addressed?

Reviewer #1: line 259: except those with detectable DNA (how does this correlate to antibody titers and any known immunodeficiency?).

line 268: how does your study prove long-lasting contact with reservoirs and why is this important?

line 283: rephrase, as this would imply that DNA detection was possible already in the 18th century.

line 326: this "potential beneficial role" should be explained and discussed in more detail in the Discussion.

Reviewer #2: Conclusions are those of a preliminary study. Cross-reactivity of the serological tests could have been addressed.

Reviewer #3: Although authors are conscious of the limitations of their study, evidence for public health relevance is missing and could benefit from additional experiments and analysis.

**Editorial and Data Presentation Modifications?**

Reviewer #1: Numerous minor grammatical errors and typos, some examples are given below.

lines 31-32: "by multiple serological and molecular testing" should go behind the bracket in line 32.

line 35: should be: sera, and: tested for

line 75: this should be "immunocompromised", I assume?

line 82: should be: Sergentomyia and perniciosus

Reviewer #2: (No Response)

Reviewer #3: (No Response)

**Summary and General Comments**

Reviewer #1: This is an interesting study with several novel aspects (detection of L. tarentolae and corresponding antibody response in humans and pets), but it could easily be condensed to a Short Report.

Reviewer #2: Line 30 – change to: Leishmania tarentolae WAS detected

Line 42 – Leishmania spp. – change accordingly throughout the manuscript

Line 52 – change to: This species, non-pathogenic to mammals, has recently been…

Line 62 – change to: With over 20 Leishmania spp. as causative agents, THE leishmaniases are neglected tropical diseases…

Line 67 – add comma: most important species of zoonotic concern, with domestic dogs

Line 68 – change to: The Mediterranean basin is A hyperendemic area

Line 70 – change to: immunocompetent hosts, ARE 5–10 times greater 

Line 77 – change to: Tyrrhenian and the low Adriatic Coasts and ON the islands, according to the distribution of the sand

Line 79 – centre

Line 81 – parasites (instead of parasite)

Line 90 – change to: In addition, this species, non-pathogenic to humans, has…

Lines 93-96 – sampling of dogs and cats should also be mentioned

Line 122 – which criteria (statistical, etc.) have determined these samples sizes?

Line 123 – change to: and n=101 from Linosa) without any specific sign of leishmaniasis, were COLLECTED and sent to the

Line 126 – were purposely collected samples (under the frame of this study, for health check analyses) all from Linosa? Please, discriminate how many were from the analyses lab and purposely collected

Lines 127-128 – which criteria (statistical, etc.) have determined these samples sizes?

Line 130 – delete “anamnestic” and change to: … sampling animals were clinically examined and data (i.e., age, sex, breed, clinical…

Line 140 – change to: L. tarentolae by IFAT, whereas CANINE and FELINE sera to both Leishmania spp. WERE TESTED only by IFAT.

Line 179 – change to: … 1 (ITS1, ~300bp) and AMPLIFICATION RUN AS DESCRIBED ELSEWHERE [26].

Line 186 – use CI (instead of CIs)

Line 199 – to both Leishmania spp.

Are there any results regarding agreement between serology and PCR for humans? The same question for dogs and cats

Line 216 – insert comma after gene,

Line 261 – Are these seronegative patients? (The detection of parasite’s DNA in few patients suggests that the parasitemia is very low or…)

Line 263 – which of the protozoa?

Line 264 – replace individuals with “humans” (in order to avoid confusion with dogs and cats)

Line 277 – could not this agreement be due to a lack of specificity of the serological tests between L. infantum and L. tarentolae?

Line 280 – Leishmania sp.

Line 281 – remove comma: … (i.e., 3.2%)

Line 284 – insert comma: … of a Brazilian mummy,

Line 291 – change to: may cause transient infections IN humans

Line 302 – delete: which had

Line 304 – change to: As far as the dog and cat populations living on the islands,

Line 307 – replace positive with “seropositive”

This seropositivity to both L. infantum and L. tarentolae should not be compared to seropositivity to only L. infantum

Line 316 – “canine leishmaniosis” – is this just disease or disease plus subclinical infection? Please report in the main text

Line 318 – the meaning of “firstly retrieved” is not clear

Line 323 – Leishmania spp.

Line 324 – replace charming with “fascinating”

Reviewer #3: Although the work and the research question are very interesting, the methods used seem insufficient to demonstrate the hypotheses that authors wish to test. The fact that in cell cultures L. tarentolae can transform into amastigotes, doesn’t imply that the parasite can effectively replicate in macrophages and sustain infection in mammals. The existing evidence points rather to the transient presence of the parasite in mammals. Without this evidence, it is very risky to suggest that the results obtained through serology are indicative of immunity, much less cross-immunity.

PLOS authors have the option to publish the peer review history of their article (what does this mean?). If published, this will include your full peer review and any attached files.

Reviewer #1: No

Reviewer #2: No

Reviewer #3: No
---

## [Decision Letter · Decision Letter 1]

16 Sep 2021

Dear Dr. Otranto,

We are pleased to inform you that your manuscript 'Leishmania tarentolae and Leishmania infantum in humans, dogs and cats in the Pelagie archipelago, southern Italy' has been provisionally accepted for publication in PLOS Neglected Tropical Diseases.

Best regards,

Yara M. Traub-Csekö

Associate Editor

Alvaro Acosta-Serrano

Deputy Editor

Reviewer's Responses to Questions

**Key Review Criteria Required for Acceptance?**

**Methods**

-Are the objectives of the study clearly articulated with a clear testable hypothesis stated?

-Is the study design appropriate to address the stated objectives?

-Is the population clearly described and appropriate for the hypothesis being tested?

-Is the sample size sufficient to ensure adequate power to address the hypothesis being tested?

-Were correct statistical analysis used to support conclusions?

-Are there concerns about ethical or regulatory requirements being met?

Reviewer #1: (No Response)

Reviewer #2: (No Response)

Reviewer #3: (No Response)

**Results**

-Does the analysis presented match the analysis plan?

-Are the results clearly and completely presented?

-Are the figures (Tables, Images) of sufficient quality for clarity?

Reviewer #1: (No Response)

Reviewer #2: (No Response)

Reviewer #3: (No Response)

**Conclusions**

-Are the conclusions supported by the data presented?

-Are the limitations of analysis clearly described?

-Do the authors discuss how these data can be helpful to advance our understanding of the topic under study?

-Is public health relevance addressed?

Reviewer #1: (No Response)

Reviewer #2: (No Response)

Reviewer #3: (No Response)

**Editorial and Data Presentation Modifications?**

Reviewer #1: (No Response)

Reviewer #2: (No Response)

Reviewer #3: (No Response)

**Summary and General Comments**

Reviewer #1: The authors have satisfyingly addressed all my comments.

Reviewer #2: (No Response)

Reviewer #3: (No Response)

PLOS authors have the option to publish the peer review history of their article (what does this mean?). If published, this will include your full peer review and any attached files.

Reviewer #1: No

Reviewer #2: No

Reviewer #3: No

---

## [Editor Report · Acceptance letter]

20 Sep 2021

Dear Prof. Otranto,

We are delighted to inform you that your manuscript, "</i>Leishmania tarentolae</i> and </i>Leishmania infantum</i> in humans, dogs and cats in the Pelagie archipelago, southern Italy," has been formally accepted for publication in PLOS Neglected Tropical Diseases.

Best regards,

Shaden Kamhawi

co-Editor-in-Chief

Paul Brindley

co-Editor-in-Chief
